# Structural analysis of the manganese transport regulator MntR from *Bacillus halodurans* in apo and manganese bound forms

Myeong Yeon Lee, Dong Won Lee, Hyun Kyu Joo, Kang Hwa Jeong, Jae Young Lee⬤ *

Department of Life Science, Dongguk University-Seoul, Ilsandong-gu, Goyang-si, Gyeonggi-do, Republic of Korea

* jylee001@dongguk.edu

**Data Availability Statement:** All relevant data are within the manuscript and its Supporting Information files. The coordinates and structure factors have been deposited in the Protein Data

## Abstract

The manganese transport regulator MntR is a metal-ion activated transcriptional repressor of manganese transporter genes to maintain manganese ion homeostasis. MntR, a member of the diphtheria toxin repressor (DtxR) family of metalloregulators, selectively responds to $Mn^{2+}$ and $Cd^{2+}$ over $Fe^{2+}$, $Co^{2+}$ and $Zn^{2+}$. The DtxR/MntR family members are well conserved transcriptional repressors that regulate the expression of metal ion uptake genes by sensing the metal ion concentration. MntR functions as a homo-dimer with one metal ion binding site per subunit. Each MntR subunit contains two domains: an N-terminal DNA binding domain, and a C-terminal dimerization domain. However, it lacks the C-terminal SH3-like domain of DtxR/IdeR. The metal ion binding site of MntR is located at the interface of the two domains, whereas the DtxR/IdeR subunit contains two metal ion binding sites, the primary and ancillary sites, separated by 9 Å. In this paper, we reported the crystal structures of the apo and $Mn^{2+}$-bound forms of MntR from *Bacillus halodurans*, and analyze the structural basis of the metal ion binding site. The crystal structure of the $Mn^{2+}$-bound form is almost identical to the apo form of MntR. In the $Mn^{2+}$-bound structure, one subunit contains a binuclear cluster of manganese ions, the A and C sites, but the other subunit forms a mononuclear complex. Structural data about MntR from *B. halodurans* supports the previous hypothesizes about manganese-specific activation mechanism of MntR homologues.

## Introduction

Metal ions are essential for living organisms because iron, zinc, and manganese ions act as cofactors for many proteins which are involved in photosynthesis, nerve transmission, and defense against toxins[1]. Manganese ions are important in many fundamental cellular processes, including protection against oxidative stress and the synthesis of the deoxyribonucleotides required for DNA replication[2,3]. However, and excess of manganese ions can be toxic [4,5]. Therefore, in order to maintain homeostasis, it is important for cells to sense and

Bank (PDB): apo BhMntR, PDB ID, 6KTA; Mn2+-bound BhMntR, PDB ID, 6KTB.

**Funding:** This work was supported by the National Research Foundation of Korea (NRF) grant funded by the Korea government (2017R1D1A1B03032109 to JYL); and the Agriculture Research Center (ARC) program of the Ministry for Food, Agriculture, Forestry and Fisheries, Korea [710013-03-1-SB120 to JYL]. The funders had no role in study design, data collection and analysis, decision to publish, or preparation of the manuscript.

**Competing interests:** The authors have declared that no competing interests exist.

respond to manganese ion concentrations[6,7]. Metalloregulatory proteins regulate metal ion homeostasis in bacteria by binding metal ions, leading to the activation or repression of the transcription of genes involved in import or efflux of the ions[8,9]. Each metalloregulatory protein has a different ligand selectivity for allosteric activation[10].

The transcriptional regulation and manganese binding of MntR from *Bacillus subtilis* has been well studied. The manganese transport regulator (MntR) functions as a homodimer and is activated by $Mn^{2+}$ to repress the expression of two manganese uptake systems, MntABCD and MntH, in response to elevated concentrations of $Mn^{2+}$[11]. Recent studies have shown that MntR activates the expression of two efflux systems, MneP and MneS, in *Bacillus subtilis*[9]. MntR is a member of the DtxR/IdeR family, which maintains iron ion homeostasis in bacteria[12]. *Corynebacterium diphtheriae* DtxR and *Mycobacterium tuberculosis* IdeRs consist of three domains: an N-terminal HTH-motif DNA binding domain (domain 1), a dimerization domain (domain 2), and a C-terminal SH3-like domain (domain 3), which is absent in MntR family proteins[13–15]. MntR consists of two domains: an N-terminal HTH-motif DNA binding domain (domain 1) and a C-terminal dimerization domain (domain 2)[5]. The DtxR/IdeR family proteins have two major metal binding sites 9.0 Å apart, called the primary and ancillary sites[7,16]. MntR is shorter than DtxR/IdeR family and the ancillary site of MntR is absent, because of the lack of an SH3-like domain in MntR[8]. The metal binding site of MntR is located between domains 1 and 2, corresponding to the primary site in DtxR/IdeR[7]. From previous structural studies it is known that the metal binding site of *B. subtilis* MntR consists of several residues including Asp8 and Glu11 in domain 1, and His77, Glu99, Glu102 and His103 in domain 2. There are two types of metal ion binding conformations in MntR, the AB conformer, and the AC conformer, resulting from differences in amino acid residues involved in metal coordination and distances between the two metal ions[5]. In the AB conformer, Asp8, Glu11, Glu102 and His103 interact with a B site $Mn^{2+}$ ion, and the metal binding sites are separated 3.3 Å. In contrast, Asp8, Glu99, Glu102 and His103 interact with a C site $Mn^{2+}$ ion, and the sites are separated 4.4 Å in the AC conformer[4].

The metal coordination geometry of MntR is essential for the generation of selective responses to cognate metals. Larger metal cations ($Mn^{2+}$ and $Cd^{2+}$) form a binuclear complex with MntR and are fully activated. However, when bound to small metal cations ($Fe^{2+}$, $Co^{2+}$, and $Zn^{2+}$), the metal ions do not fully occupy the site, but form a mononuclear complex, resulting in low activity[17].

The crystal structures of the MntR family have been determined from several bacterial species, including *Bacillus subtilis*[18], *Escherichia coli*[7], and *Mycobacterium tuberculosis*[17]. Previous structural studies of the MntR family have described how conformation changes depending on whether the sites are bound to cognate metal ions, and how such conformational changes induce a dissociation of cognate DNA from the MntR protein[19]. The MntR homologue (BH2807, *Bh*MntR) in *Bacillus halodurans* is a protein consisting of 139 amino acids, and has 78% sequence identity with MntR from *B.subtilis* (*Bs*MntR). Further sequence comparisons of *B. halodurans* MntR show that it is 31% identical to *E. coli* MntR, 26% identical to *M. tuberculosis* MntR, 24% identical to *C. diphtheriae* DtxR, and 26% identical to *T. acidophilum* IdeR (Fig 1A).

Although the crystal structures of MntR from bacterial species have been determined, the metal coordination and selectivity are not fully understood. To further understand the metal binding site of the MntR protein, we determined crystal structures of the apo and manganese-bound forms of MntR from *B. halodurans*. The structures revealed that *Bh*MntR forms a binuclear complex with manganese ions in the AC conformer.

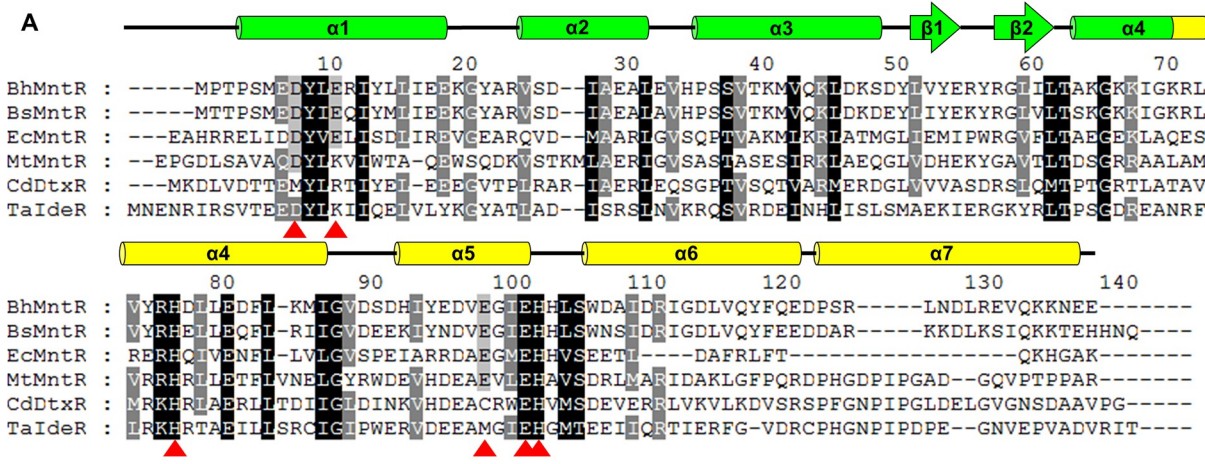

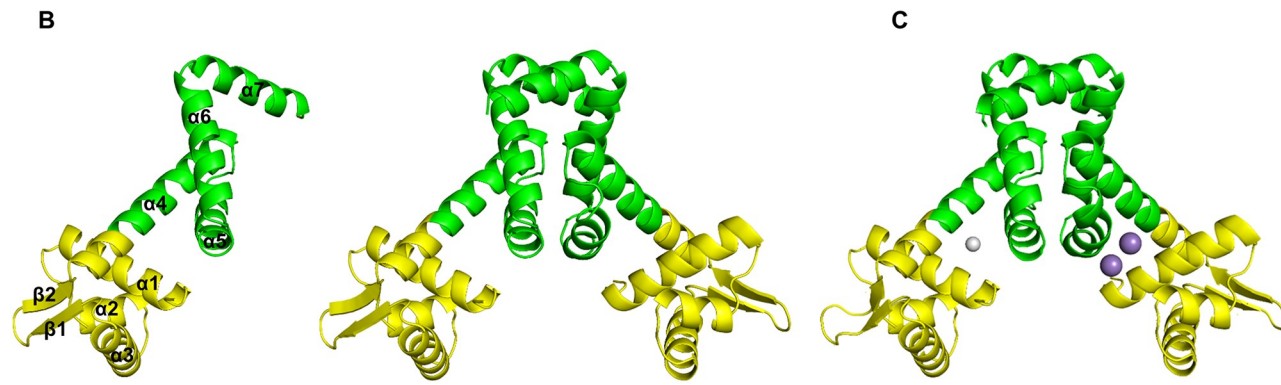

**Fig 1. Multiple sequence alignment and overall structure of *Bh*MntR.** (A) Multiple sequence alignment of *Bh*MntR with other MntR homologues. The secondary structures of *Bh*MntR are indicated above the sequence. The highly conserved and partially conserved residues are shaded in black and gray boxes, respectively. The residues involved in metal binding are shown as red triangles at the bottom of the sequence. (B) The monomeric and dimeric structures of apo *Bh*MntR. *Bh*MntR are composed of an N-terminal DNA binding domain (yellow) and a C-terminal domain (green). (C) The dimeric structure of $Mn^{2+}$-bound form *Bh*MntR. One subunit contained binuclear manganese ions (purple), while the other subunit forms a mononuclear complex with magnesium ion (gray).

## Materials and methods

### Expression and purification of *Bh*MntR

The *mntR* genes were amplified using polymerase chain reaction (PCR) using the genomic DNA of *B. halodurans* as a template. The amplified *mntR* genes were inserted into an NdeI/XhoI-digested vector pET-28b(+) (Novagen, Germany) producing a hexahistidine-tag (His-tag) at its N-terminus. The recombinant *Bh*MntR was transformed and expressed in *E.coli* BL21(DE3) Star pLysS cells (Invitrogen, USA). The transformed cells were grown at 310 K to an $OD_{600}$ of ~0.5 in Luria-Bertani medium supplemented with 30 μg $mL^{-1}$ kanamycin and chloramphenicol. Overexpression of recombinant BhMntR was induced with 1.0 mM isopropyl $\beta$-D-1-thiogalactopyranoside (IPTG) and allowed to grow for four hours at 303 K. The cells were harvested by centrifugation at 4,200 *g* for 15 minutes at 277 K and frozen immediately at 193 K. The cell pellets (6~8 g) were resuspended in buffer A (20 mM Tris-HCl pH 8.0, 0.5 M NaCl, and 10%(*v/v*) glycerol) containing 1 mM phenylmethylsulfonyl fluoride and homogenized using an ultrasonic processor (Sonics & Materials™, Vibra Cell VCX 750, USA).

The insoluble fraction was removed by centrifugation at 28,000 g (Supra 22 K; Hanil BioMed Inc., Korea) for one hour at 277 K.

The recombinant *Bh*MntR in the soluble fraction was loaded on a nickel-charged His-trap immobilized metal affinity chromatography (IMAC) column (GE Healthcare, UK) pre-equilibrated with buffer A, washed with buffer A containing 60 mM imidazole, and eluted from the column with buffer B (20 mM Tris-HCl pH 8.0, 0.5 M NaCl, 10%(*v/v*) glycerol, and 300 mM imidazole) to immobilized-metal-affinity-chromatography (IMAC) on a Ni-NTA resin (GE Healthcare). The BhMntR was further purified by size exclusion chromatography using a Superdex 200 gel-filtration column (GE Healthcare, UK), employing with elution buffer (20 mM Tris-HCl pH 8.0, 0.2 M NaCl, 5% (*v/v*) glycerol, 1 mM dithiothreitol (DTT), and 2 mM MgCl$_2$). The purity of *Bh*MntR was assessed using 12% (*v/v*) SDS-PAGE. The purified *Bh*MntR was concentrated to 17 mg/ml using centrifugal filter units (Millipore) and aliquots of the protein were stored at 193 K.

### Crystallization and X-ray diffraction data collection

Crystallization of *Bh*MntR was performed using the sitting-drop vapor diffusion method at 296 K with 96-well crystallization plates (SWISSCI MRC, UK) and commercial screening solution from Anatrace, Hampton Research, Emerald Biosystems and Molecular Dimensions. Each sitting-drop was prepared by mixing 0.75 μl of the concentrated protein and the reservoir solution. The crystals of apo *Bh*MntR were grown in reservoir solution containing 0.1 M sodium/potassium phosphate pH 6.2 and 0.4 M magnesium formate. The co-crystallization of BhMntR with manganese ions was unsuccessful. Mn$^{2+}$-bound crystals were obtained by soaking with 50 mM MnCl$_2$ for one hour in apo crystals, grown in 0.1 M sodium phosphate pH 6.5 and 0.4 M magnesium formate.

Each crystal was transferred into a cryo-protectant solution containing the reservoir solution with 20%(*v/v*) glycerol and flash-cooled in a liquid nitrogen stream. X-ray diffraction data were collected at 100 K with a Pilatus3 6M detector using synchrotron radiation on a Beamline 11C of the Pohang Accelerator Laboratory (PAL) in Korea. The crystals were exposed to X-rays for 1.0 second per image, and 180 frames were obtained for each 1.0˚ oscillation. All data were processed and scaled using *DENZO* and *SCALEPACK* from the *HKL-2000* program suite [20]. The detailed data collection statistics are summarized in Table 1.

### Structure determination and refinement

The structure of apo *Bh*MntR was solved by molecular replacement using the program *PHASER MR* from the CCP4 program suite[21] using the apo *Bs*MntR structure (PDB code 2HYG) [18] as a search model. The initial model was further improved by alternating cycles of manual building using the *COOT* program[22], and the model was refined with the *PHENIX* program package[21]. The refined model was evaluated using *MolProbity*[23]. The refinement statistics of apo *Bh*MntR and Mn$^{2+}$ bound *Bh*MntR are presented in Table 1.

## Results and discussion

### Model building and quality

The apo crystal structure of *Bh*MntR was determined at 2.3 Å resolution using molecular replacement with the MntR model of *B. subtilis* (2HYG). The structure was refined to crystallographic R$_{work}$ and R$_{free}$ values of 18.9% and 22.9%, respectively with good geometry. The refined model (PDB code 6KTA) contained two *Bh*MntR subunits which formed a homodimer, four molecules of glycerol, and 135 water molecules in the asymmetric unit. The model

**Table 1. Data collection and refinement statistics.**

|  | Apo *Bh*MntR | Mn²⁺-bound *Bh*MntR |
|---|---|---|
| **Data collection** |  |  |
| Space group | P2₁2₁2₁ | P2₁2₁2₁ |
| Unit-cell parameters |  |  |
| $a$, $b$, $c$ (Å) | 39.66, 89.20, 109.76 | 39.53, 89.37, 109.95 |
| $\alpha$, $\beta$, $\gamma$ (°) | 90.00, 90.00, 90.00 | 90.00, 90.00, 90.00 |
| Wavelength (Å) | 0.97941 | 0.97960 |
| Resolution (Å) | 50.00–2.30 (2.34–2.30) | 50.00–2.50 (2.54–2.50) |
| Number of observations | 115,264 | 93,537 |
| Unique reflections | 17,942 | 14,131 |
| Data completeness (%) | 99.1 (99.2) | 99.9 (99.9) |
| Redundancy | 6.4 (6.0) | 6.6 (7.1) |
| Averge I/σ(I) | 16.2 (4.5) | 14.1 (7.9) |
| $R_{merge}$ (%)[a] | 11.9 (36.4) | 12.8 (36.7) |
| **Refinement statistics** |  |  |
| Resolution (Å) | 46.74–2.30 | 37.20–2.50 |
| $R_{work}$ / $R_{free}$ (%) | 18.9/22.9 | 17.1/21.8 |
| No. of non-H atoms | 2439 | 2379 |
| Protein | 2280 | 2284 |
| Ligands | 24 | 18 |
| Water | 135 | 77 |
| rmsd bonds (Å) | 0.008 | 0.008 |
| rmsd angles (°) | 0.938 | 0.930 |
| Average B-factor | 35.7 | 27.9 |
| Protein | 28.48 | 21.65 |
| Ligands | 49.57 (glycerol) | 47.20 (Mn²⁺,Mg²⁺,Phosphate) MGGMGD |
| Water | 38.97 | 30.07 |
| Ramachandran plot (%) |  |  |
| Favored | 98.53 | 98.53 |
| Allowed | 1.47 | 1.47 |
| Outliers | 0 | 0 |

[a]$R_{merge} = \Sigma_h\Sigma_i|I(h)_i - <I(h)>|/\Sigma_h\Sigma_iI(h)_i$, where $I(h)$ is the intensity of reflection $h$, $\Sigma_h$ is the sum over all reflections, and $\Sigma_i$ is the sum over i measurements of reflection $h$.

was validated using *MolProbity*[23]. The C-terminal region of chains A (residue 139) and B (residues 136–139) were poorly ordered, due to lack of electron-density maps. Mn²⁺-bound *Bh*MntR crystals were obtained by soaking with 50 mM MnCl₂ for one hour in apo crystals. The crystal structure of the Mn²⁺-bound form was determined at 2.5 Å resolution, and the binuclear manganese ions were clearly evident in the 2Fo-Fc map and omit maps, whereas a magnesium ion was observed in the other subunit. The structure of Mn²⁺-bound *Bh*MntR was refined with a crystallographic R_work value of 17.1% and an R_free value of 21.8%. Each subunit of the Mn²⁺-bound *Bh*MntR was well defined, except for the C-terminal residue 139. The refined model (PDB code 6KTB) contains two *Bh*MntR subunits, three molecules of phosphate, and 77 water molecules in the asymmetric unit. All refined models for *Bh*MntR showed favored or allowed regions in a Ramachandran plot.

## Overall structure of *B. halodurans* MntR

Each *Bh*MntR subunit was composed of seven α-helices and two β-strands, which could be divided into an N-terminal Helix-Turn-Helix (HTH) DNA binding domain (domain1, residues 1–71) and a C-terminal dimerization domain (domain2, residues 72–139) (Fig 1). The N and C-terminal domains were connected by a long linker helix (α4) that extended from the wing to the dimer interface. The *Bh*MntR was a homodimeric structure, with approximate dimensions of 40Å × 55Å × 80Å. The N-terminal DNA binding domain consisted of three α-helices and two strands of antiparallel β-sheet, forming a winged HTH motif that putatively interacted with DNA. Because helix α3 of the HTH motif could be responsible for DNA recognition, we speculate that the positively charged residues (Lys41, Lys45, and Lys48) in helix α3 are involved in DNA binding. Domain 2, the dimerization domain, is composed of four α-helices (α4–α7). Domains 1 and 2 are connected by the long linker helix α4 (residues 64–87).

The two subunits form a dimeric structure, related by a non-crystallographic 2-fold axis (Fig 1). The buried surface area of the dimer is about 1300 Å$^2$, approximately 14% of the monomer surface area. The dimeric *Bh*MntR is stabilized by the hydrogen bonds and hydrophobic interactions along helices α4 to α7; 14 residues were involved in hydrophobic interactions and eight residues in hydrogen bonds. (PDBePISA protein–protein interaction server: http://www.ebi.ac.uk/msd-srv/prot_int/ and PDBsum generate: http://www.ebi.ac.uk/thornton-srv/databases/pdbsum/Generate.html). The dimer interface is mainly produced by hydrophobic side chains such as Phe83, Ile87, Gly88, Val 89, Gly100, Ile101, Leu105, Ala109, Ile113, Leu116, Tyr119, Phe120, Leu130, and Val133. Ten hydrogen bonds were formed between Asp90 N and Asp108 Oδ2, between Asp97 Oδ1 and Ser106 N, between Asp97 Oδ2 and Ser106 Oγ, between Tyr119 Oη and Tyr119 Oη, between Glu122 Oε1 and Lys136 Nζ, between Asp115 Oδ1 and Asn137 Nδ2, and between Gln118 Oε1 and Asn137 Nδ2. This finding demonstrated that *Bh*MntR exists as a functional dimer in solution. Two subunits in the asymmetric unit of *Bh*MntR showed little structural difference, with a root-mean-square deviation (r.m.s.d.) value of 1.31 Å for 137 Cα atoms in residues 1–137 (S1 Fig). There were few structural differences between apo and Mn$^{2+}$-bound dimeric forms, with a r.m.s.d. value of 0.49 Å for 276 Cα atoms.

## Metal binding site

We obtained Mn$^{2+}$-bound crystals by soaking with 50 mM MnCl$_2$ in apo crystals, and confirmed using an omit map and an anomalous map showing two peaks at the counter levels even at 5σ (Fig 2A and S2 Fig). The metal binding site appeared to be fully occupied in one subunit with the temperature factors for the two manganese ions being 51.02 Å$^2$ and 63.88 Å$^2$, respectively. However, the other subunit contained a magnesium ion which was coordinated by the side chains of Glu99, Glu102, and two water molecules (Fig 2C). The two manganese ions were found at the interface between the HTH domain and the dimerization domain and formed a binuclear complex separated by 4.5 Å, labeled as the A and C sites (AC conformer).

The binuclear manganese ions were liganded by six amino acid residues: Asp8 and Glu11 contributed by domain 1, and His 77, Glu99, Glu102, and His103 contributed by domain 2 (Fig 2B). The two manganese ions (Mn$_A$ and Mn$_C$) were jointly coordinated by the carboxylate oxygens of Glu99 and Glu102 from domain 2. Each metal ion was individually coordinated by Glu11 (Mn$_A$), His77 (Mn$_A$), His103 (Mn$_C$) and Asp8 (Mn$_C$). The Mn$_A$ ion was coordinated by seven atoms: Glu11 Oε1/Oε2, His77 Nδ1, Glu99 Oε2, Glu102 Oε1/Oε2, and Wat95 O. In addition, the His77 Nε2 made a hydrogen bond with Glu81 Oε1, while the His77 Nδ1 in the other subunit made a hydrogen bond with Glu81 Oε1 via a water molecule (Fig 2C). The Mn$_c$ ion was coordinated by five atoms: Asp8 Oδ1, Glu99 Oε1, Glu102 Oε2,

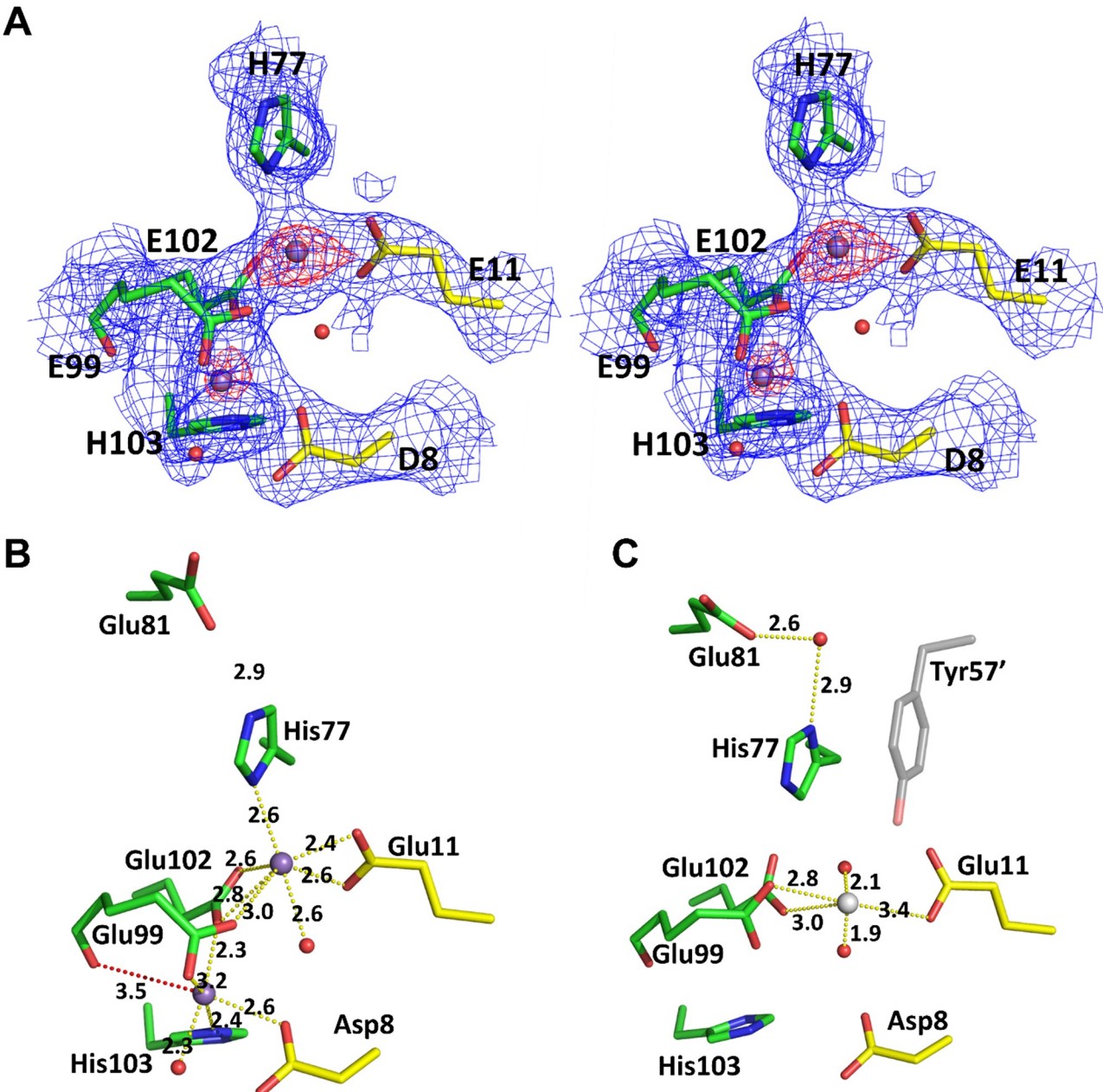

**Fig 2. Metal ion binding site in the *B. halodurans* MntR.** (A) Stereoview of metal binding site in the *B. halodurans* MntR. A σ$_A$-weighted electron density map (2Fo-Fc map) contoured at 1.0σ (blue). Omit map was calculated, contoured at 3σ (red). The Mn$^{2+}$ atoms (purple) are depicted with surrounding residues (yellow sticks from domain1 and green sticks from domain2). (B) Metal binding site with binuclear manganese ions. The coordination with binuclear manganese ions and the distance between Mn$_c$ and the backbone carbonyl oxygen of Glu99 are shown in yellow and red, respectively. (C) Metal binding site with a magnesium ion (gray). Unlike binuclear manganese ions binding site, a magnesium ion was coordinated by the side chains of Glu99, Glu102, and two water molecules. The His77 made a hydrogen bond with Glu81 via a water. The symmetry-related Tyr57 is colored gray.

His103 Nε2, and Wat37 O, while the C site of *Bs*MntR has octahedral coordination geometry. In the Mn$^{2+}$-bound *Bs*MntR structure, the backbone carbonyl oxygen of Glu99 coordinated with the Mn$_c$ ion, but this interaction between them was too distant to interact in the *Bh*MntR, at 3.5 Å (Fig 2B).

In the other subunit of the $Mn^{2+}$-bound MntR structure, no manganese binding was observed, although the residues are positioned appropriately to form a manganese binding site. The reason for the lack of bound manganese ions at this site is unclear. The side chain of His77, which is strictly conserved in the MntR/IdeR family, had a different rotamer with a hydrogen bond via a water molecule to Glu81 and was also stabilized by π-π interaction with symmetry-related Tyr57 (Fig 2C). These interactions could block the proper rotamer of His77 to coordinate with $Mn_A$ ion in this subunit. These findings suggested that the His77 flip in *Bh*MntR could initiate metal binding in the presence of manganese ions. It will be valuable to verify the role of His77 at the metal binding site in the future experiments.

## Structural comparison to other MntR homologue

We carried out structural and sequence comparisons among DtxR/MntR proteins from various organisms using the *Clustal Omega*[24] and DALI server[25]. The best five matches were those of the metal-dependent DtxR/MntR family. They were (1) the manganese transport regulator, MntR from *B. subtilis*[4] (PDB code 2F5F; r.m.s. d. of 1.2 Å for 137 equivalent Cα positions in residue 2–138 of *Bh*MntR, a Z-score of 19.1, and a sequence identity of 78%), (2) the MntR from *E. coli*[7] (PDB code 2H09; r.m.s.d. of 2.1 Å for 118 equivalent Cα positions in residue 1–114 and 116–119 of *Bh*MntR, a Z-score of 15.7, and a sequence identity of 35%), (3) the *C. diphtheriae* DtxR in complex with DNA[26] (PDB code 1BI2; r.m.s. d. of 2.2 Å for 119 equivalent Cα positions in residue 1–119 of *Bh*MntR, a Z-score of 14.1, and a sequence identity of 26%), (4) the *M. tuberculosis* IdeR in complex with DNA[27] (PDB code 1U8R; r.m.s.d. of 1.8 Å for 116 equivalent Cα positions in residue 3–119 of *Bh*MntR, a Z-score of 13.8, and a sequence identity of 28%), and (5) the *T. acidophilum* IdeR in complex with DNA[28] (PDB code 4O6J; r.m.s.d. of 2.5 Å for 114 equivalent Cα positions in residue 4–118 of *Bh*MntR, a Z-score of 12.7, and a sequence identity of 29%).

Previous studies revealed that *Bs*MntR shows conformational changes when bound to the manganese ions by inducing a hinge bending motion between residues 72 and 75[18]. To investigate the hinge motion properties of *Bh*MntR, we compared the domain orientation, by superimposing the Cα atoms of domain 2 (72–139) in the *Bh*MntR structure with those of the apo *Bs*MntR (PDB code 2HYG), the $Mn^{2+}$-bound *Bs*MntR (PDB code 2F5D), and the $Zn^{2+}$-bound *Bs*MntR (PDB code 2EV6). The r.m.s deviations in Cα positions for domain 2 (residues 72–139) are 0.90 Å, 0.76 Å and 0.95 Å (S1 Table). When the dimerization domain is superimposed, the DNA binding domains varies by 2.4–8.5 Å at residue Lys41. The movement of the DNA binding domain with respect to domain 2 is centered at residue Tyr75 of helix α4, and is tilted by 4.5–17° (Fig 3A). There is no loss of hydrogen bonding within helix α4 upon metal binding, while hydrogen bonding was lost within helix α4 in *T. acidophilum* IdeR. When measured between the Cα atoms of Lys41, at the center of helix α3, the domain separation of apo and manganese bound *Bh*MntR are 37.4 Å and 37.5 Å, respectively, while the distance between the Lys41 in apo, $Mn^{2+}$-bound, and $Zn^{2+}$-bound *Bs*MntR are 39.2 Å, 32.1Å and 30.7Å, respectively (Fig 3B). There was little domain movement between apo and $Mn^{2+}$-bound *Bh*MntR, possibly due to crystal packing or the presence of positively charged ions of $Na^+$ (~0.5 M) and $Mg^{2+}$ (~0.4 M) during the crystallization process. It will be important to verify the domain movement upon metal binding by co-cystallization in future experiments.

## Conclusions

We reported the crystal structures of *Bh*MntR: apo, and $Mn^{2+}$-bound forms. Our results showed that *Bh*MntR is composed of two distinct domains in the homodimeric form, and its overall structure is similar to those of other MntR homologues. The two manganese ions

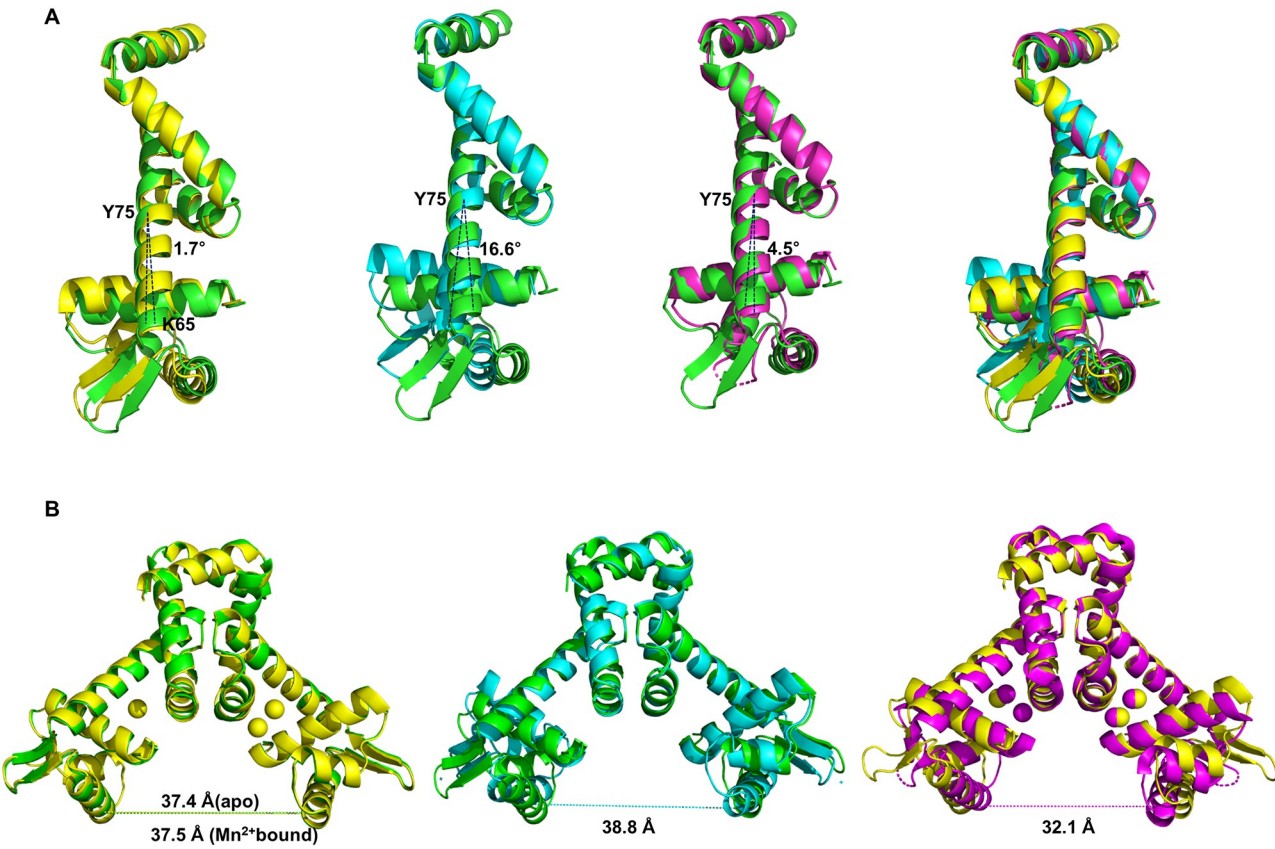

**Fig 3. Structural comparison between *Bh*MntR and *Bs*MntR.** (A) Superimposing apo *Bh*MntR with Mn²⁺-bound *Bh*MntR and *Bs*MntR. Superimposition was based on the dimerization domain of one subunit. The angle was centered at residue Tyr75 of the helix α4 and measured between the residues of Lys65. The apo *Bh*MntR, Mn²⁺-bound *Bh*MntR, apo *Bs*MntR and Mn²⁺-bound *Bs*MntR are indicated in green, yellow, cyan and magenta, respectively. (B) Distances of residue Lys41 in the dimeric structure. The dimeric structures of MntR are aligned by the dimerization domain.

formed a binuclear cluster in the metal binding site of *Bh*MntR, via six amino acid residues; three strictly conserved residues (His77, Glu102 and His103) in the IdeR/MntR family, two residues (Asp8 and Glu99) conserved in the MntR family, and a Glu11 conserved in MntR from *B. subtilis* and *E. coli*. The manganese ion in A site was liganded with heptageometry as shown in *Bs*MntR, whereas the manganese ion in the C site was incompletely liganded with five atoms. The sixth atom, the carbonyl oxygen of Glu102, was too far away to coordinate with the $Mn_C$ ion. Therefore, *Bh*MntR did not cause movement of the domain to bind DNA upon manganese ion binding. Binuclear metal ions were not formed in the other subunit due to the crystal packing and the flipping of His77. The side chain of His77 was flipped and stabilized by hydrogen bonding and hydrophobic stacking. In order to initiate metal binding, the side chain of His77 was flipped to interact with the carboxylate of Glu81. Although the functional assignment of metal binding site for *Bh*MntR is tentative, this structural model is applicable to other MntR homologous structures.

## Supporting information

**S1 Fig. R.m.s.d plot of *Bh*MntR.**
(TIF)

**S2 Fig. Anomalous maps in metal ion binding site of *Bh*MntR.** (A) Stereoview of metal binding site with binuclear manganese ions of the $Mn^{2+}$-bound *Bh*MntR. A $\sigma_A$-weighted electron density map (2Fo-Fc map) contoured at 1.0σ (blue). Anomalous map was calculated, contoured at 2σ (red). The $Mn^{2+}$ atoms (purple) are depicted with surrounding residues (yellow sticks from domain1 and green sticks from domain2). (B) Stereoview of metal binding site with a magnesium ion in other subunit of the $Mn^{2+}$-bound *Bh*MntR. A $\sigma_A$-weighted electron density map (2Fo-Fc map) contoured at 1.0σ (blue). Anomalous map was calculated, contoured at 2σ (red). (C) Anomalous maps were calculated around metal binding site with binuclear manganese ions with different contour level (5σ, 4σ, and 3σ).
(TIF)

**S1 Table. Structural comparisons of *Bh*MntR with *Bs*MntR.**
(DOCX)

**S1 File. Apo *Bh*MntR coordinate.**
(PDB)

**S2 File. Apo *Bh*MntR structure factor.**
(MTZ)

**S3 File. Mn-bound *Bh*MntR coordinate.**
(PDB)

**S4 File. Mn-bound *Bh*MntR structure factor.**
(MTZ)

**S5 File. Validation report of apo *Bh*MntR structure.**
(PDF)

**S6 File. Validation report of Mn-bound *Bh*MntR structure.**
(PDF)

## Acknowledgments

We thank the staff members at the Pohang Accelerator Laboratory beamline 5C and 11C for their help with data collection.

## Author Contributions

**Conceptualization:** Hyun Kyu Joo, Jae Young Lee.

**Data curation:** Myeong Yeon Lee, Dong Won Lee, Hyun Kyu Joo, Kang Hwa Jeong.

**Formal analysis:** Myeong Yeon Lee.

**Investigation:** Kang Hwa Jeong.

**Supervision:** Jae Young Lee.

**Writing – original draft:** Myeong Yeon Lee, Dong Won Lee, Jae Young Lee.

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
