## [Decision Letter · Decision Letter 0]

11 Oct 2019

PONE-D-19-24432

Structural analysis of the manganese transport regulator MntR from Bacillus halodurans in apo and manganese bound forms

PLOS ONE

Dear Dr. Jae Young Lee

Thank you for submitting your manuscript to PLOS ONE. After careful consideration, we feel that it has merit but does not fully meet PLOS ONE’s publication criteria as it currently stands. Therefore, we invite you to submit a revised version of the manuscript that addresses the points raised during the review process.

ACADEMIC EDITOR: Please try to improve your manuscript according to the reviewers' criticism. 

We would appreciate receiving your revised manuscript by Nov 25 2019 11:59PM. To enhance the reproducibility of your results, we recommend that if applicable you deposit your laboratory protocols in protocols.io, where a protocol can be assigned its own identifier (DOI) such that it can be cited independently in the future. For instructions see: http://journals.plos.org/plosone/s/submission-guidelines#loc-laboratory-protocols

We look forward to receiving your revised manuscript.

Kind regards,

Eugene A. Permyakov, Ph.D., Dr.Sci.

Academic Editor

**PLOS ONE**

**Journal Requirements:**

2. We  found  text overlap between the current submission and your previously published works outside the Method section:

     - https://www.nature.com/articles/s41598-018-31676-z

    - http://scripts.iucr.org/cgi-bin/paper?S1399004714004118

    - http://koreascience.or.kr/article/JAKO201425560113942.page

In your revision ensure you cite all your sources (including your own works), and quote or rephrase any duplicated text outside the methods section. Further consideration is dependent on these concerns being addressed."

**Comments to the Author**

1. Is the manuscript technically sound, and do the data support the conclusions?

Reviewer #1: Yes

Reviewer #2: Partly

2. Has the statistical analysis been performed appropriately and rigorously? 

Reviewer #1: Yes

Reviewer #2: N/A

3. Have the authors made all data underlying the findings in their manuscript fully available?

Reviewer #1: Yes

Reviewer #2: Yes

4. Is the manuscript presented in an intelligible fashion and written in standard English?

Reviewer #1: Yes

Reviewer #2: Yes

5. Review Comments to the Author

Reviewer #1: This paper presents crystal structure data on a further member of the MntR family of magnesium transporter regulators. As expected, the structure is largely similar to previously reported structures. But there are some new details here regarding metal binding sites. The crystallographic work appears to be mainly in order, but with one issue noted below. I have one substantive critique of the analysis and some relatively minor comments and suggestions prior to publication.

Important concern:

Nothing is said by the authors about anomalous scattering. They might have generated stronger anomalous signal from the Mn had they collected data at a longer x-ray wavelength, and in view of this it’s not clear why the wavelength was chosen to be at 0.979 Ang (which is a good choice for selenium studies, not applicable here). Maybe the authors could get lucky and anomalous signal in a map would be detectable even at the sub-optimal wavelength where they collected the data (f’’ for Mn looks to be about 1.3 electrons at 0.979 Ang). The authors should calculate an anomalous difference map to see if the binuclear site shows up. If it does, then they could go on to look at the other, supposedly unoccupied, site. There they inferred a Mg ion, but as noted above it doesn’t make a lot of sense that a Mg would be there when it wasn’t there in the apo form. It’s conceivable that the density in the other site is a partially occupied Mn (not Mg). And if so maybe an anomalous map would give an indication. Having a better assignment of what’s in the other site would improve the findings of the paper.

Minor points:

1) There are some places where an editing for English usage will be important (e.g. recombination vs recombinant).

2) Lines 61-69 in the Introduction discuss some structural details of the MntR protein. As it is written is sounds like the authors might be describing results from the present study (which would be out of place in the Introduction), whereas they are presumably discussing what is known about the structure from prior work. This should be made clearer, e.g. “From previous structural studies it is understood that…”

3) The R-values are reported with a number of significant digits that seems one too many.

4) The legend to Fig 2 needs to explain better the distinction (in panels b AND c) between the site where the binuclear site is seen vs the manganese site.

5) The authors show 2Fo-Fc and omit (difference) maps to show the metal sites. This is ok. But they have the opportunity here to also calculate and show maps that are based on differences between the Fobs from the two data sets (i.e. Fobs(+metal) – Fobs(-metal), phased with a model without metals). This kind of map is a more direct examination of the observed differences and less model-dependent. The authors should examine this kind of map. Presumably it will show up the binuclear Mn cluster just as the Fo-Fc map did. But the authors might discern more about the unexpected magnesium site. Currently they say that no magnesium is seen in the ‘apo’ structure, despite the high concentration of Mg in the crystallization conditions. If Mg is really absent from that site in the apo form, then they should see a positive peak there in the Fo(+metal) – Fo(-metal) map.

Reviewer #2: This manuscript describes MntR (from Bacillus halodurans) structures in the apo and metal bound states. The assignment of Mn2+ and Mg2+ in the binding sites was not solid. It needs to be substantialized by additional measurements such as X-ray anomalous signals. The reported structures do not provide new biological insights in addition to the many existing MntR structures.

6. PLOS authors have the option to publish the peer review history of their article (what does this mean?). If published, this will include your full peer review and any attached files.

Reviewer #1: No

Reviewer #2: No

---

## [Author Response · Author response to Decision Letter 0]

18 Oct 2019

Dear Eugene A. Permyakov, Ph.D. Academic Editor, PLOS ONE

Reference code PONE-D-19-24432

TITLE: “Structural analysis of the manganese transport regulator MntR from Bacillus halodurans in apo and manganese bound forms”

Thank you very much for your kindly informing me of the reviewer’s comments on my manuscript submitted for a possible publication in the PLOS ONE. Please find the uploaded revised manuscript at the web address.

The revised manuscript takes into account all the comments made by the reviewers and editor. A separate list is attached to this letter to describe all the changes in detail.

Sincerely yours,

List of changes by reviewer 1

1. << Important concern: Nothing is said by the authors about anomalous scattering. They might have generated stronger anomalous signal from the Mn had they collected data at a longer x-ray wavelength, and in view of this it’s not clear why the wavelength was chosen to be at 0.979 Ang (which is a good choice for selenium studies, not applicable here). Maybe the authors could get lucky and anomalous signal in a map would be detectable even at the sub-optimal wavelength where they collected the data (f’’ for Mn looks to be about 1.3 electrons at 0.979 Ang). The authors should calculate an anomalous difference map to see if the binuclear site shows up. If it does, then they could go on to look at the other, supposedly unoccupied, site. There they inferred a Mg ion, but as noted above it doesn’t make a lot of sense that a Mg would be there when it wasn’t there in the apo form. It’s conceivable that the density in the other site is a partially occupied Mn (not Mg). And if so maybe an anomalous map would give an indication. Having a better assignment of what’s in the other site would improve the findings of the paper. >> - We calculated an anomalous difference map with Mn2+-bound BhMntR data and found the two clear peaks showing the manganese binuclear ions in metal binding site. The additional information was added on lines 200-202, page 10, stating that “We obtained Mn2+-bound crystals by soaking with 50 mM MnCl2 in apo crystals, and confirmed using an omit map and an anomalous map showing two peaks at the counter levels even at 5σ (Fig 2A and S2 Fig).”. In addition, the requested anomalous maps of Mn2+-bound BhMntR structures was added to supplemental figure (S2 Fig). 

2. << There are some places where an editing for English usage will be important (e.g. recombination vs recombinant). >> - The English spelling was carefully checked and corrected on line 104, page 5, stating that “recombinant”.

3. << Lines 61-69 in the Introduction discuss some structural details of the MntR protein. As it is written is sounds like the authors might be describing results from the present study (which would be out of place in the Introduction), whereas they are presumably discussing what is known about the structure from prior work. This should be made clearer, e.g. “From previous structural studies it is understood that…” >> - The requested information was added as recommended on lines 62-69, page 3, stating that “From previous structural studies it is known that the metal binding site of B. subtilis MntR consists of several residues including Asp8 and Glu11 in domain 1, and His77, Glu99, Glu102 and His103 in domain 2.” 

4. << The R-values are reported with a number of significant digits that seems one too many. >> - The R-values were corrected as recommended in text and table. 

5. << The legend to Fig 2 needs to explain better the distinction (in panels b AND c) between the site where the binuclear site is seen vs the manganese site. >> - The original Fig2 legend sentence was changed on lines 212-217, page 10, stating that “(B) Metal binding site with binuclear manganese ions. The coordination with binuclear manganese ions and the distance between Mnc and the backbone carbonyl oxygen of Glu99 are shown in yellow and red, respectively. (C) Metal binding site with a magnesium ion. Unlike binuclear manganese ions binding, the magnesium ion (gray) forms a mononuclear cluster and the His77 made a hydrogen bond with Glu81 via a water (red).” 

6. << The authors show 2Fo-Fc and omit (difference) maps to show the metal sites. This is ok. But they have the opportunity here to also calculate and show maps that are based on differences between the Fobs from the two data sets (i.e. Fobs(+metal) – Fobs(-metal), phased with a model without metals). This kind of map is a more direct examination of the observed differences and less model-dependent. The authors should examine this kind of map. Presumably it will show up the binuclear Mn cluster just as the Fo-Fc map did. But the authors might discern more about the unexpected magnesium site. Currently they say that no magnesium is seen in the ‘apo’ structure, despite the high concentration of Mg in the crystallization conditions. If Mg is really absent from that site in the apo form, then they should see a positive peak there in the Fo(+metal) – Fo(-metal) map.>> - We also calculated the Fo(+metal) – Fo(-metal) map. The two positive peaks were clearly shown around metal binding site indicating the binuclear manganese ions but no clear map was shown in magnesium binding site in the other subunit. In addition, the anomalous maps were calculated and shown in Supporting Information (S2 Fig). 

List of changes by reviewer 2

1. << This manuscript describes MntR (from Bacillus halodurans) structures in the apo and metal bound states. The assignment of Mn2+ and Mg2+ in the binding sites was not solid. It needs to be substantialized by additional measurements such as X-ray anomalous signals. The reported structures do not provide new biological insights in addition to the many existing MntR structures. >> - We calculated an anomalous difference map with Mn2+-bound BhMntR data and found the two clear peaks showing the binuclear manganese ions in metal binding site. The additional information was added on lines 200-202, page 10, stating that “We obtained Mn2+-bound crystals by soaking with 50 mM MnCl2 in apo crystals, and confirmed using an omit map and an anomalous map showing two peaks at the counter levels even at 5σ (Fig 2A and S2 Fig).”. In addition, the requested anomalous maps of BhMntR structures was added to Supplementary figure (S2 Fig). 

Journal Requirements:

1. << When submitting your revision, we need you to address these additional requirements.

http://www.journals.plos.org/plosone/s/file?id=wjVg/PLOSOne_formatting_sample_main_body.pdf and http://www.journals.plos.org/plosone/s/file?id=ba62/PLOSOne_formatting_sample_title_authors_affiliations.pdf >>. Our revised manuscript was generated with PLOS ONE style templates as requested.

2. << We found text overlap between the current submission and your previously published works outside the Method section:

 - https://www.nature.com/articles/s41598-018-31676-z

 - http://scripts.iucr.org/cgi-bin/paper?S1399004714004118

 - http://koreascience.or.kr/article/JAKO201425560113942.page

 In your revision ensure you cite all your sources (including your own works), and quote or rephrase any duplicated text outside the methods section. Further consideration is dependent on these concerns being addressed." >> - We carefully checked text overlaps between the current submission and your previously published works outside the Method section. Main overlaps were found in Acknowledgement Section and the other minor overlap was rephrased on lines 291-293, page 13, stating that “Although the functional assignment of metal binding site for BhMntR is tentative, this structural model is applicable to other MntR homologous structures.” 

3. << We note that you have stated that you will provide repository information for your data at acceptance. Should your manuscript be accepted for publication, we will hold it until you provide the relevant accession numbers or DOIs necessary to access your data. If you wish to make changes to your Data Availability statement, please describe these changes in your cover letter and we will update your Data Availability statement to reflect the information you provide. >> - We already provided the accession numbers (6KTA and 6KTB) for PDB coordinates and structure factors and those data were included in Supporting Information.

---

## [Editor Report · Decision Letter 1]

21 Oct 2019

Structural analysis of the manganese transport regulator MntR from Bacillus halodurans in apo and manganese bound forms

PONE-D-19-24432R1

Dear Dr. Jae Young Lee,

We are pleased to inform you that your manuscript has been judged scientifically suitable for publication and will be formally accepted for publication once it complies with all outstanding technical requirements.

With kind regards,

Eugene A. Permyakov, Ph.D., Dr.Sci.

Academic Editor

PLOS ONE
---

## [Editor Report · Acceptance letter]

24 Oct 2019

PONE-D-19-24432R1 

Structural analysis of the manganese transport regulator MntR from *Bacillus halodurans* in apo and manganese bound forms 

Dear Dr. Lee:

I am pleased to inform you that your manuscript has been deemed suitable for publication in PLOS ONE. Congratulations! Your manuscript is now with our production department. 

With kind regards,

on behalf of

Prof. Eugene A. Permyakov 

Academic Editor

PLOS ONE